# Oncometabolite D-2-Hydroxyglutarate enhances gene silencing through inhibition of specific H3K36 histone demethylases

Ryan Janke[1,2], Anthony T Iavarone[2], Jasper Rine[1,2]*

[1]Department of Molecular and Cell Biology, University of California, Berkeley, Berkeley, United States; [2]California Institute for Quantitative Biosciences, University of California, Berkeley, Berkeley, United States

**Abstract** Certain mutations affecting central metabolism cause accumulation of the oncometabolite D-2-hydroxyglutarate which promotes progression of certain tumors. High levels of D-2-hydroxyglutarate inhibit the TET family of DNA demethylases and Jumonji family of histone demethylases and cause epigenetic changes that lead to altered gene expression. The link between inhibition of DNA demethylation and changes in expression is strong in some cancers, but not in others. To determine whether D-2-hydroxyglutarate can affect gene expression through inhibiting histone demethylases, orthologous mutations to those known to cause accumulation of D-2-hydroxyglutarate in tumors were generated in *Saccharomyces cerevisiae*, which has histone demethylases but not DNA methylases or demethylases. Accumulation of D-2-hydroxyglutarate caused inhibition of several histone demethylases. Inhibition of two of the demethylases that act specifically on histone H3K36me2,3 led to enhanced gene silencing. These observations pinpointed a new mechanism by which this oncometabolite can alter gene expression, perhaps repressing critical inhibitors of proliferation.

**\*For correspondence:** jrine@berkeley.edu

**Competing interests:** The authors declare that no competing interests exist.

## Introduction

Among the biggest surprises from cancer genome sequencing has been the extent to which mutations affecting metabolic enzymes appear as drivers of specific types of cancers. Isocitrate dehydrogenases (IDH), enzymes that convert isocitrate to α-ketoglutarate, are prime examples. Specific IDH mutations are highly prevalent features of certain subsets of cancers, including 60–90% of gliomas and secondary glioblastomas, 20% of late-stage acute myelogenous leukemia, ~50% of central and periosteal cartilagenous tumors, and 10–20% of interhepatic cholangiocarcinoma (*Parsons et al., 2008*; *Molenaar et al., 2014*). The high frequency and patterns of IDH mutations suggest that they are important in early tumor development. In the case of acute myelogenous leukemia, IDH mutations also contribute to progression of these tumors (*Molenaar et al., 2014*; *Reitman and Yan, 2010*). With rare exception, IDH mutations in tumors occur exclusively as heterozygous missense mutations. The most frequently documented IDH mutations occur as substitutions of arginine 132 in IDH1 or the equivalent arginine (172) in IDH2. The specific amino acid substitution varies across and within tumor subtypes.

Tumor-associated IDH mutations appear to abolish the conversion of isocitrate to α-ketoglutarate, and instead promote a partial reverse reaction in which α-ketoglutarate is reduced to form D-2-hydroxyglutarate (D2-HG) (*Dang et al., 2009*; *Rendina et al., 2013*). D2-HG is a metabolite that is typically maintained at low levels in human cells through the activity of the enzyme D-2-

hydroxyglutarate dehydrogenase (D2HGDH) (*Achouri et al., 2004*). Cellular D2-HG levels are commonly elevated in IDH-mutant tumor cells, sometimes to millimolar levels (*Gross et al., 2010*; *Dang et al., 2009*).

The structural similarity between D2-HG and α-ketoglutarate inspire models for D2-HG's role in tumor formation centered on antagonism of α-ketoglutarate-dependent enzymes. Indeed, the α-ketoglutarate-dependent TET family of DNA demethylases and Jumonji histone demethylases are inhibited by high levels of D2-HG in vitro and in cell culture, resulting in DNA and histone hypermethylation and accompanied by changes in gene expression (*Figueroa et al., 2010*; *Xu et al., 2011*; *Chowdhury et al., 2011*; *Lu et al., 2012*). There is strong circumstantial evidence from acute myeloid leukemias (AMLs) that DNA demethylases are the targets of the D2-HG produced by the IDH mutations: AMLs tend to have either the mutations in IDH or mutations in the genes encoding TET enzymes, but not both (*Figueroa et al., 2010*; *Gaidzik et al., 2012*). In the case of gliomas and other types of tumors, there is no such simple dichotomy, suggesting the possibility of other important targets of D2-HG in these tumors such as histone demethylases. Indeed elevated bulk histone methylation has been reported for some tumors producing D2-HG, but whether critical changes in gene expression result from inhibition of DNA demethylation or inhibition of histone demethylation in these tumors is an important and unresolved question (*Chowdhury et al., 2011*; *Gaidzik et al., 2012*; *Lu et al., 2012*). Further, tumors with elevated D2-HG levels show bulk increases in histone methyl marks typically associated with both transcriptional activation (H3K4, H3K36) and repression (H3K9, H3K27) (*Turcan et al., 2012*; *Lu et al., 2012*). The increase of global histone methylation marks could conceivably lead to architectural changes in chromatin that alter heterochromatin/euchromatin states, or change the capacity to initiate and undergo transcription, or some combination of both. Which processes underlie the critical transcriptional changes that promote development and progression in D2-HG-accumulating tumors is unknown. Here, the causality of transcriptional changes due to elevated D2-HG was tested in *Saccharomyces cerevisiae*. Elevated D2-HG levels led to increased gene repression at a heterochromatic locus. The mechanism behind this effect on transcription was pinpointed to elevated H3K36 methylation as a result of D2-HG inhibition of Jumonji-domain-containing demethylases Rph1 and Gis1.

## Results

To study the impact of IDH mutations on gene expression, an analogous tumor-associated IDH mutation was made in *Saccharomyces cerevisiae* (*IDP2-R132H*) at the *IDP2* locus. The yeast and human proteins share 61% amino acid identity (*Figure 1—figure supplement 1*). This mutation is analogous to the most common mutation found in low-grade gliomas and secondary glioblastomas in the human ortholog *IDH1* (reviewed in *Waitkus et al., 2016*). Heterochromatic gene silencing at the *HML* locus was used as the assay for detecting changes in gene expression of a locus for which the transcriptional state is epigenetically inherited using the Cre-Reported Altered States of Heterochromatin (CRASH) assay. In this assay, loss of silencing allows for transient expression of a cre-recombinase gene inserted within *HML*, leading to a permanent and heritable switch from RFP expression to GFP expression (*Figure 1A*) (*Dodson and Rine, 2015*). The frequency of switches from RFP to GFP expression within a colony serves as the readout of the strength/stability of gene silencing in heterochromatin at *HML*. A decrease in gene silencing leads to more cells switching from RFP to GFP. Enhanced gene silencing leads to a reduction in switches from RFP to GFP.

### Mutations expected to produce D2-HG resulted in enhanced gene silencing

Because *IDP2* is expressed only on non-fermentable carbon sources (*Haselbeck and McAlister-Henn, 1993*), cells were plated on solid medium containing glycerol as the sole carbon source to induce *IDP2* expression. The frequency of loss-of-silencing events, detected as green spots or sectors within a colony, was quantified using MORPHE (*Liu et al., 2016*). The *IDP2-R132H* mutation increased the stability of heterochromatin (p<0.0001; Student's *t* test) compared to wild-type *IDP2*, resulting in fewer switches from RFP to GFP expression (*Figure 1B*). This effect was dependent on *IDP2* expression, as no effect was observed when cells were plated on medium containing glucose, which represses *IDP2* expression (*Figure 1—figure supplement 2*). Deletion of *IDP2* did not affect the stability of heterochromatin suggesting that *IDP2-R132H* results in a gain-of-function. These

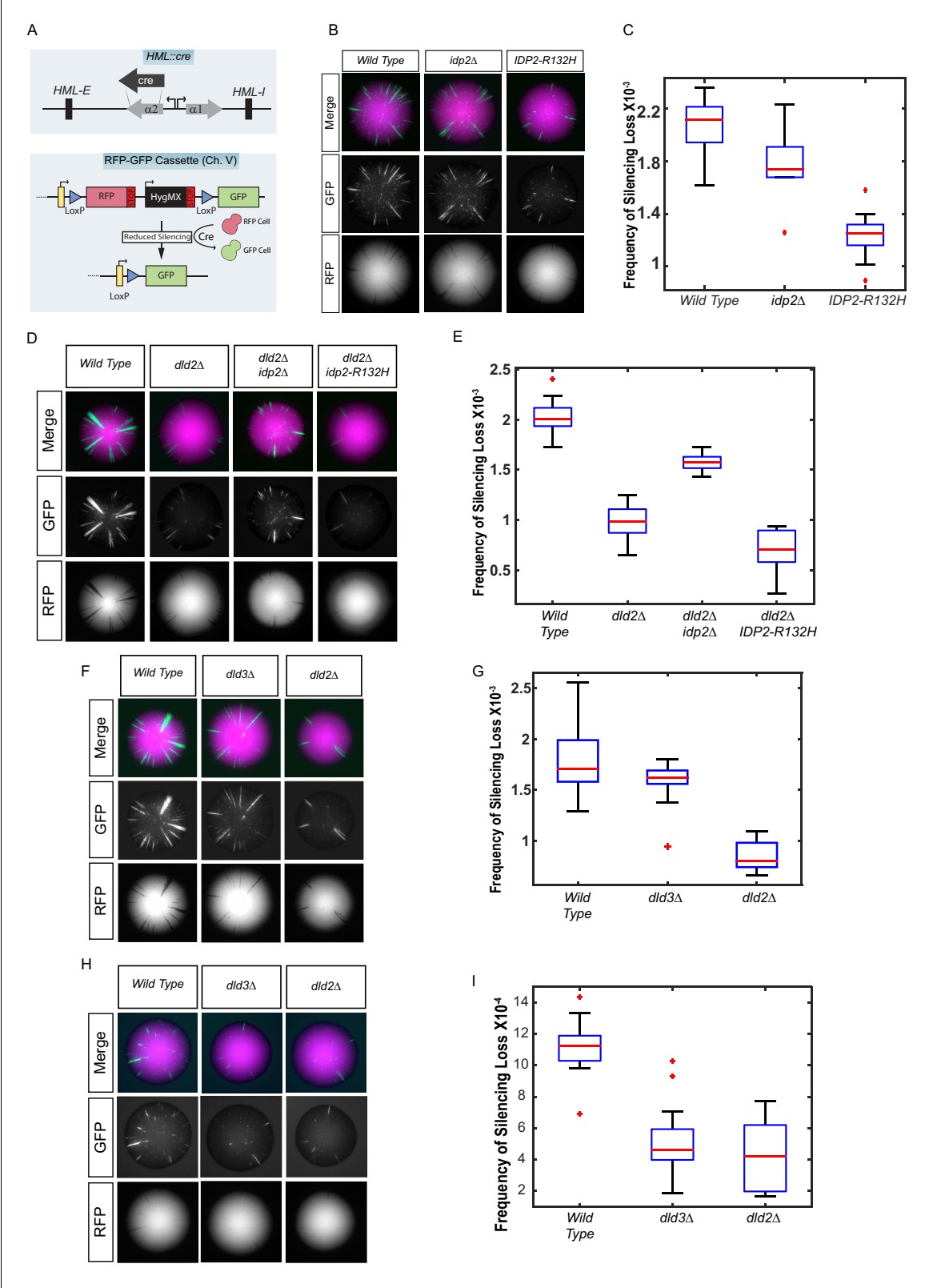

**Figure 1.** Yeast versions of tumor-associated isocitrate dehydrogenase mutation caused increased stabilization of heterochromatin. (A) Illustration of the CRASH (Cre reported altered states of heterochromatin) assay used to measure the strength of heterochromatic gene silencing at the *HML* locus. (B) Images of colonies with the CRASH reporter assay that were wild type (JRY10791), *idp2Δ* (JRY10732), or *IDP2-R132H* (JRY10731). (C) Box plots of the average frequency of loss-of-silencing events from mutants in panel B calculated using MORPHE (*Liu et al., 2016*). (D) Representative colony images of *Figure 1 continued on next page*

*Figure 1 continued*

CRASH reporter strains with wild type (JRY10790) or mutant *dld2Δ* (JRY10733), *dld2Δ idp2Δ* (JRY10735), or *dld2Δ IDP2-R132H* (JRY10734). (E) The frequency of silencing loss from mutants in panel D. (F) Representative colony images of CRASH reporter strains with wild type (JRY10790) or mutant *dld3Δ* (JRY10752) and *dld2Δ* (JRY10733). (G) Plots of the frequency of silencing loss from mutants in panel F. (H) Images of colonies with the CRASH reporter assay that were wild type (JRY10790), *dld2Δ* (JRY10733), or *dld3Δ* (JRY10752). Colonies were grown on CSM-Trp-glucose agar plates for 7 days at 30°C. Representative images show the merged and separate GFP and RFP channels. (I) Plots of the frequency of silencing loss from mutants in panel H.

The following figure supplements are available for figure 1:

**Figure supplement 1.** Multiple peptide sequence alignment of human and *Saccharomyces cerevisiae* NADP+-dependent isocitrate dehydrogenases.

**Figure supplement 2.** Images of colonies with the CRASH reporter assay and *idp2Δ* or *IDP2-R132H* mutations grown on glucose.

results demonstrate that tumor-associated IDH mutations led to increased stability of gene silencing in a heterochromatic domain in *Saccharomyces.*

If *IDP2-R132H* stabilized heterochromatin through increased D2-HG levels, then other genetic changes that independently increase D2-HG levels should do the same. In human cells, D2-HG is metabolized by D2HGDH (*Achouri et al., 2004*). Loss of D2HGDH causes D2-HG to accumulate to high levels. The high degree of amino acid sequence similarity between human D2HGDH and yeast Dld2 and Dld3 suggested that both were potential yeast orthologs of human D2HGDH, as recently confirmed (*Becker-Kettern et al., 2016*). Indeed, on medium with glycerol as the sole carbon source, when oxidative phosphorylation is the primary means of ATP production, the *dld2Δ* mutant phenocopied the effect of the *IDP2-R132H* mutant and increased gene silencing compared to wild type (p<0.0001; Student's *t* test) (*Figure 1D,E*). Moreover, the *IDP2-R132H dld2Δ* double mutant resulted in a further increase in heterochromatin stability compared to wild type (p<0.0001; Student's *t* test) and *dld2Δ* alone (p<0.05; Student's *t* test) (*Figure 1D,E*). The stabilizing effect of a *dld2Δ* mutation was slightly diminished in combination with an *idp2Δ* deletion, suggesting that even wild-type Idp2 may contribute to the formation of D2-HG, either by directly producing D2-HG or indirectly by producing α-ketoglutarate as a substrate used by other enzymes for D2-HG formation. A *dld3Δ* mutation did not increase heterochromatin stability when cells were grown on glycerol medium (*Figure 1F,G*). However, when cells were grown on glucose, where ATP is primarily generated through sugar fermentation, both *dld2Δ* and *dld3Δ* mutations led to increased heterochromatin stability (*Figure 1H,I*). The increase in heterochromatin stability in the *dld2Δ* and *dld3Δ* mutants implied that D2-HG is normally produced in yeast under these and other conditions (see also *Becker-Kettern et al., 2016*).

If the increased gene silencing in the *IDP2-R132H* mutant were due to increased D2-HG levels, then the phenotype of the mutant should be dominant to wild-type *IDP2*. Conversely, if the effects on gene silencing were due to a reduced ability to convert isocitrate into α-ketoglutarate, the *IDP2-R132H* allele would likely be recessive to wild type. In an *IPD2-R132H/IDP2* heterozygous diploid strain, gene silencing was increased compared to the equivalent strain with two copies of wild-type *IDP2* (*Figure 2A,B*), demonstrating that the yeast mutation, analogous to the cancer-driver mutation in human *IDH1*, was dominant to wild-type *IDP2*. Dominance of the *IDP2-R132H* allele was observed both in wild-type *DLD2* and *dld2Δ* mutant backgrounds. Interestingly, a strain homozygous for *IDP2-R132H* had less stable gene silencing than the heterozygous strain, suggesting that the effect of *IDP2-R132H* on silencing was synergistic with a wild-type copy of *IDP2*. These results echoed the observation in $IDH1^{R132H/WT}$ tumor cell lines that knockout of the wild-type copy of *IDH1* ($IDH1^{R132H/-}$) results in decreased D2-HG formation relative to $IDH1^{R132H/WT}$ cells (*Jin et al., 2013*).

## Elevated D2-HG caused enhanced gene silencing

To establish whether both *IDP2-R132H* and *dld*-mutant strains accumulate D2-HG, metabolite extracts were prepared from cells harboring these mutations and analyzed by quantitative liquid chromatography-mass spectrometry (LC-MS). D2-HG levels in wild-type strains ranged between 10 and 60 μM, in agreement with previous levels measured in yeast (*Becker-Kettern et al., 2016*).

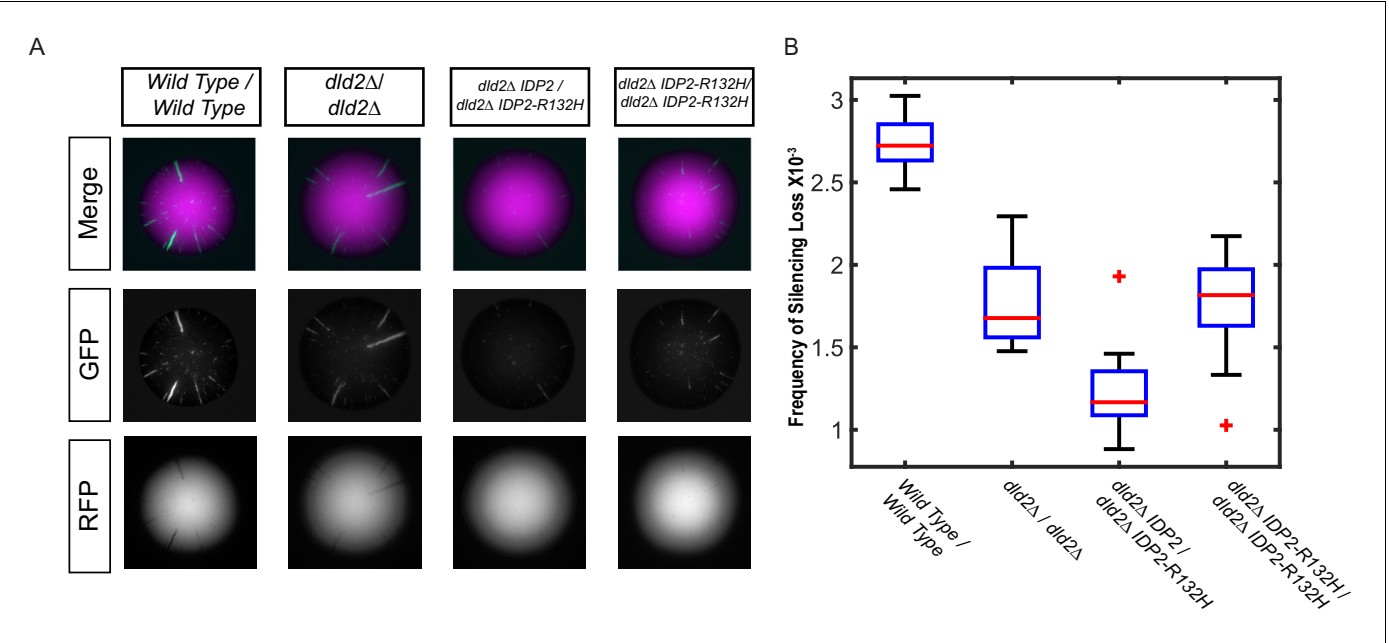

**Figure 2.** The silencing phenotype of *IDP2-R132H* was dominant. (**A**) Images of diploid colonies with the CRASH reporter assay. The diploid strains have *CRE* inserted at both copies of *HML* and two copies of the CRASH RFP-GFP reporter cassette and were wild type (JRY10750), *dld2Δ/dld2Δ* (JRY10749), *dld2Δ/dld2Δ IDP2/IDP2-R132H* (JRY10751), or *dld2Δ/dld2Δ IDP2-R132H/IDP2-R132H* (JRY10748). Cells were plated on CSM-Trp glycerol agar plates and imaged after 7 days. (**B**) Plots of the frequency of silencing loss from mutants in panel A were calculated using MORPHE.

Consistent with a role in metabolizing D2-HG, a *dld2Δ* mutation led to a ~50 fold increase in cellular D2-HG levels in cells grown on glycerol (*Figure 3A*) and a fivefold increase in cells grown on glucose (*Figure 3D*). A *dld3Δ* mutation also led to a sevenfold increase in cellular D2-HG, but this increase was observed only in cells grown on glucose (*Figure 3C,D*). Thus, D2-HG is a normal feature of the *S. cerevisiae* metabolome and is efficiently metabolized by Dld2 and Dld3, as noted (*Becker-Kettern et al., 2016*). However, Dld3 activity toward D2-HG was limited to cells grown on fermentable carbon sources like glucose. Although the increased D2-HG level in the Idp2-R132H mutant inferred from the silencing phenotype was not detected in the LC-MS analysis of cells that were able to metabolize it (*Figure 3A*), the Idp2-R132H mutation clearly contributed to D2-HG production when measured in combination with *dld2Δ* and wild-type Idp2 protein (*Figure 3B*). Tumors with the equivalent isocitrate dehydrogenase mutation (*IDH1-R132H*) are exclusively heterozygous, retaining a wild-type copy of *IDH1,* and knock-down of wild-type IDH1 reduces D2-HG produced by IDH1-R132H (*Ward et al., 2013*). Because IDH is typically a homodimer, presumably the wild-type enzyme provides a local source of α-ketoglutarate for the mutant enzyme to convert to D2-HG (*Ward et al., 2013*; *Jin et al., 2013*). To test whether D2-HG production by yeast Idp2-R132H was likewise enhanced by the presence of wild-type Idp2, cells were transformed with a plasmid containing a copy of the wild-type *IDP2* gene expressed from its native promoter. The presence of an extra copy of wild-type *IDP2* had no effect on D2-HG levels in wild-type *IDP2* cells (*Figure 3B*, bar 3 versus bar 4). However, D2-HG levels significantly increased (p<0.05) in mutant *IDP2-R132H* cells when an extra copy of wild-type *IDP2* was present (*Figure 3B*, bar 5 versus bar 4). In the same strains, the increased levels of D2-HG corresponded to an increase in silencing (*Figure 3—figure supplement 1*). An extra copy of wild-type *IDP2* had no effect on silencing in wild-type *IDP2* cells. In parallel to the results measuring D2-HG levels, an extra copy of wild-type *IDP2* enhanced silencing in mutant *IDP2-R132H* cells compared to strains with only wild-type *IDP2* (p<0.0001; Student's *t* test). These results demonstrated a strong correlation between D2-HG levels and increased silencing at a heterochromatic locus.

As a direct test of whether increased levels of D2-HG were responsible for the increase in gene silencing, cells were grown in the presence or absence of cell membrane-permeable octyl-D2-HG.

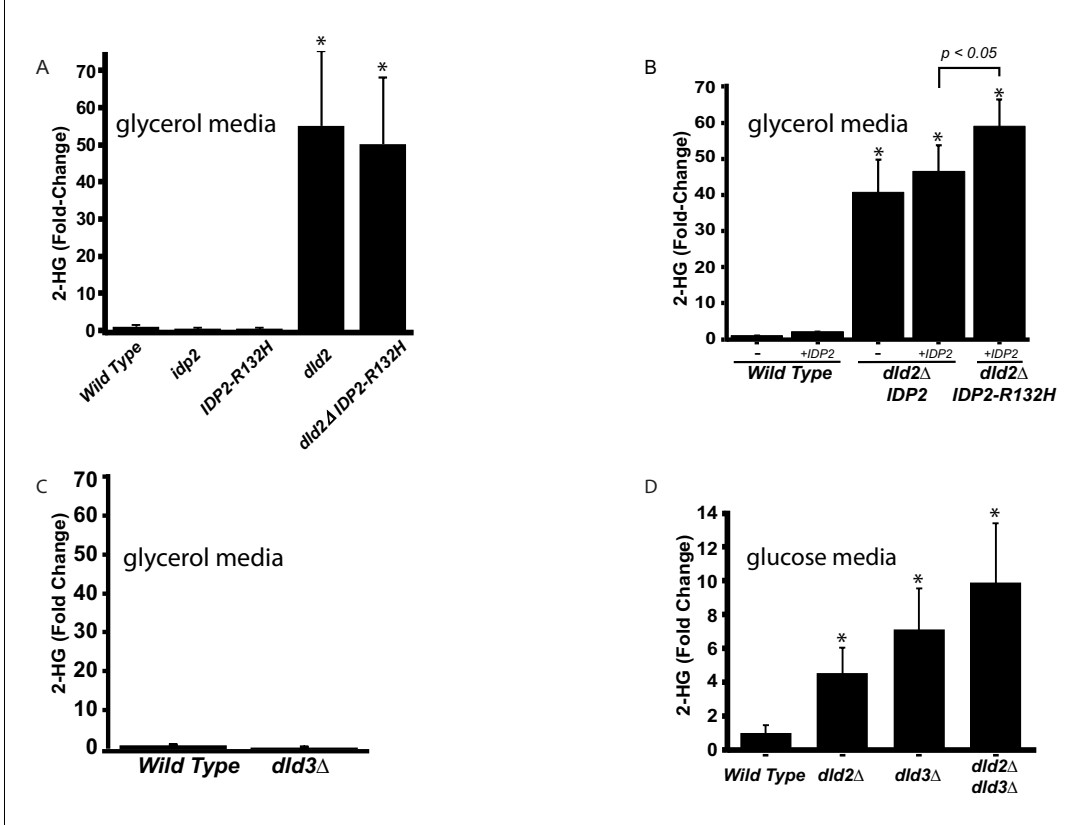

**Figure 3.** D2-HG levels increase in *dld-* and *idp* mutants. (A) Plot of the average fold change from wild type (JRY10790) in levels of D-2-hydroxyglutarate from metabolite extracts of *dld2Δ* (JRY10733), *idp2Δ* (JRY10732), *IDP2-R132H* (JRY10731), and *dld2Δ IDP2-R132H* (JRY10734) mutant strains measured by LC-mass spectrometry. (B) Plot of the average fold change from wild type in levels of D-2-hydroxyglutarate from metabolite extracts from wild type (JRY10790), *dld2Δ* (JRY10733) and *dld2Δ IDP2-R132H* (JRY10734) cells transformed with the empty vector pRS315 (denoted as -) or plasmid pJR3399 containing a wild-type copy of *IDP2* (denoted as +IDP2). (C) The average D-2-hydroxyglutarate levels from wild type (JRY10790) and *dld3Δ* (JRY10752) strains were measured and plotted as in part A. (D) The average D-2-hydroxyglutarate levels from wild type (JRY10790), *dld2Δ* (JRY10733), *dld3Δ* (JRY10752), and *dld2Δ dld3Δ* (JRY10753) mutants grown in minimal medium with 2% glucose were measured as in part A. For all panels, statistical analysis was performed using an unpaired, two-tailed (Student's) *t* test. Error bars show the standard error of the mean. Bars marked by an asterisk (*) were statistically significantly different (p<0.05) from wild type.

The following figure supplement is available for figure 3:

**Figure supplement 1.** Increased D2-HG levels correspond to an increase in silencing.

Treatment with octyl-D2-HG had no impact on silencing in wild-type cells that readily metabolized D2-HG (*Figure 4A*). However, treatment of *dld2Δ* mutant cells with octyl-D2-HG resulted in increased gene silencing compared to the untreated cells (p<0.05) (*Figure 4B*) establishing that increases in D2-HG were causal for stabilizing gene silencing.

## D2-HG did not enhance a locus-specific gene repression mechanism

In principle, D2-HG accumulation could impact other forms of gene repression. The *GAL1* promoter is repressed in media containing glucose. By placing the *cre* gene under control of the *GAL1* promoter, it was possible to measure the impact of D2-HG on another type of transcriptional repression. When *dld2Δ* mutations were introduced into the *pGAL1::cre* strain, repression of *cre* was maintained to the same degree as in wild type, indicating that the impact of D2-HG did not extend to all forms of repression (*Figure 5*).

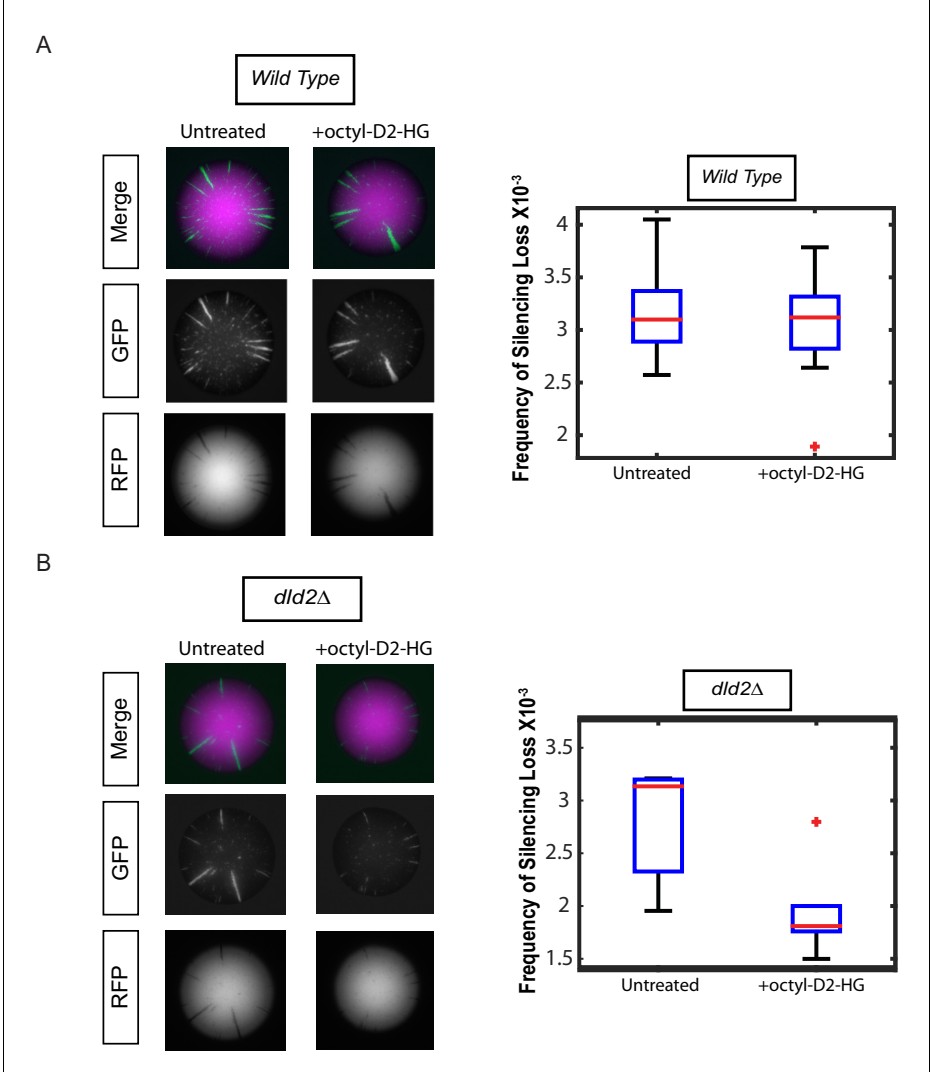

**Figure 4.** Treatment with cell-permeable octyl-D2-HG increased heterochromatin silencing. (**A**) Images of wild type (JRY10790) CRASH assay reporter colonies that were grown on CSM-Trp 3% Glycerol (untreated) or with 100 µM octyl-D2-HG added to the medium (Left Panel). Plots of the frequency of silencing loss calculated using MORPHE (Right Panel). (**B**) Images of *dld2Δ* (JRY10733) CRASH assay reporter colonies that were grown on CSM-Trp 3% Glycerol (untreated) or with 100 µM octyl-D2-HG added to the medium (Left Panel). Plots of the frequency of silencing loss from colonies calculated using MORPHE. The data in *Figure 4A and B* were from independent experiments conducted on different days with different batches of media. MORPHE-based comparisons should be restricted to experiments performed in parallel at the same time (*Liu et al., 2016*).

## D2-HG inhibited multiple histone demethylases

D2-HG can inhibit 2-oxoglutarate/Fe(II)-dependent dioxygenases. This class of enzymes includes TET DNA demethylases and Jumonji-C class of histone demethylases, which are both important epigenetic regulators previously implicated in progression of certain cancers. Since both families of demethylase are inhibited by D2-HG (*Xu et al., 2011*; *Figueroa et al., 2010*; *Chowdhury et al., 2011*; *Lu et al., 2012*), we tested whether the enhanced gene silencing caused by D2-HG in yeast was due to inhibition of a demethylase activity. DNA methylation does not occur in *S. cerevisiae,* but histone methylation impacts heterochromatin at *HML* (*Osborne et al., 2009*; *van Leeuwen et al., 2002*). To determine if increased D2-HG lead to increased bulk histone methylation, or whether the effects were localized to particular methylated species, histone methylation levels were analyzed from yeast

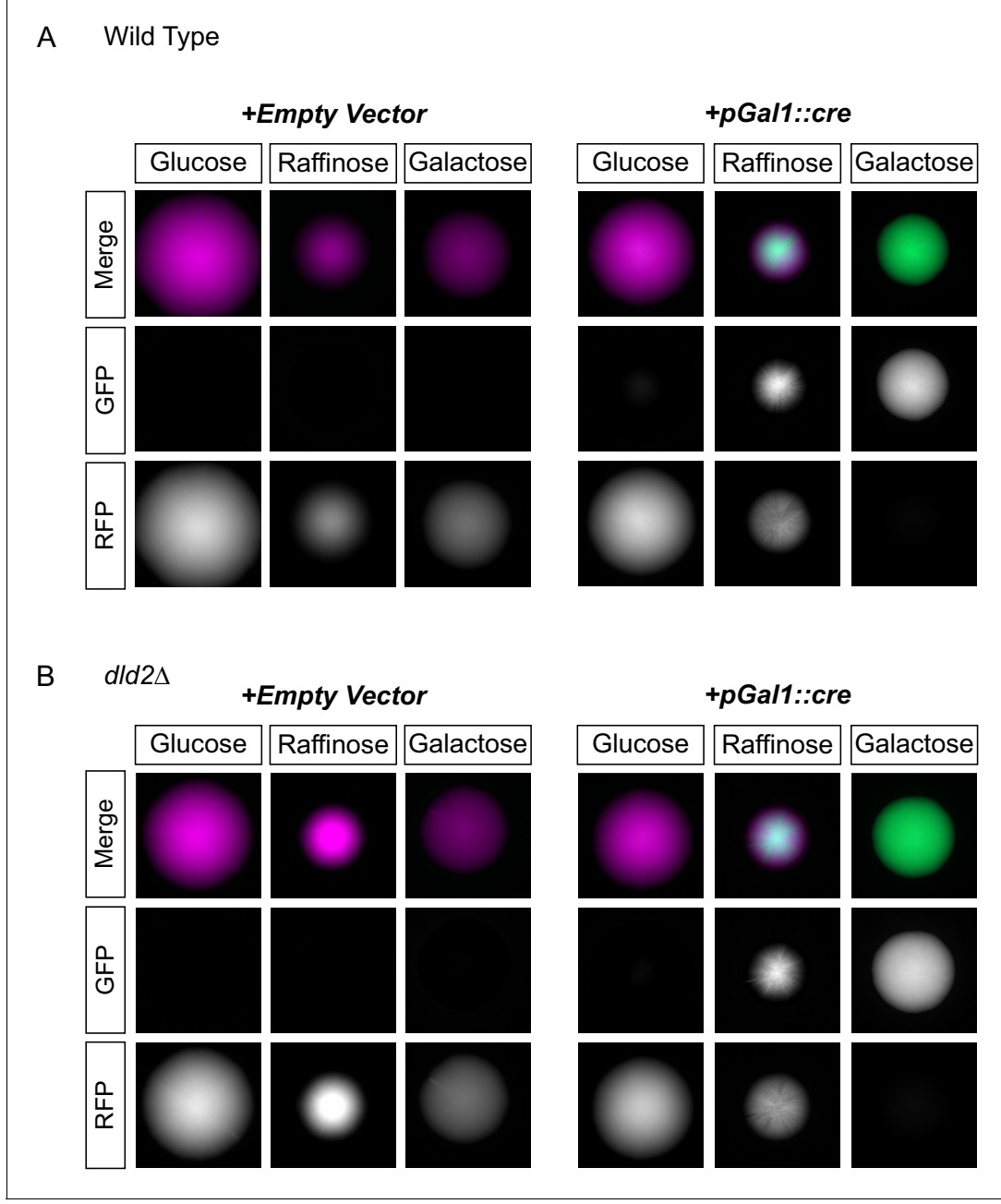

**Figure 5.** D2HG did not enhance a locus-specific gene repression mechanism. (**A**) Images of colonies of CRASH reporter strains with wild-type *HML* (JRY10757) and either a pRS413 empty vector (left panels) or a vector with *cre* expressed from the *GAL1* promoter (pSH62) (right panels). Cells were plated on CSM-His-Trp with the indicated carbon source and grown for 7 days at 30°C. Representative images show the merged and separate GFP and RFP channels. (**B**) The same experiment as in part A was performed in a strain containing a *dld2* deletion (JRY10758).

extracts prepared from either wild type or *dld2Δ IDP2-R132H* mutant cells using antibodies against specific histone methyl marks. Bulk levels of H3K36-mono and -trimethyl and H3K4-trimethyl marks were significantly higher (p<0.01) in *dld2Δ IDP2-R132H* strains compared to wild type (**Figure 6A and B**, **Figure 6—figure supplement 1**). The antibody for the H3K36 dimethyl species exhibited too much background to determine the statistical significance of the apparent increase in the mutant cells. Thus, bulk levels of all three methylated species of H3K36 methylation and H3K4 methylation increased in strains that accumulated high levels of D2-HG, consistent with D2-HG inhibition of

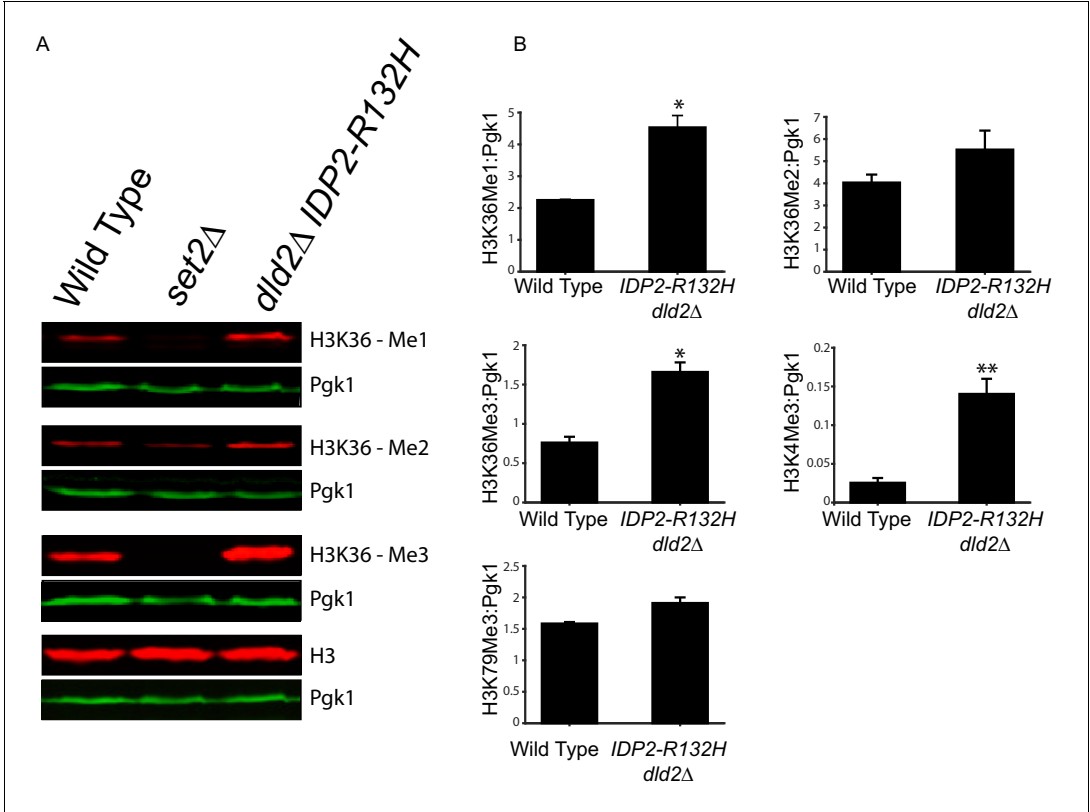

**Figure 6.** Bulk histone methylation increased in strains with high levels of D2-HG. (A) Immuno-blot analysis of H3 histone methylation states from wild type (JRY10790), set2Δ (JRY10746), and dld2Δ IDP2-R132H (JRY10734) mutants. The upper panel of each set shows immuno-blot signals generated from antibodies against mono-, di-, or trimethyl H3K36, and total H3. The lower panel of each set shows a loading control immuno-blot signal generated from an antibody against Pgk1 from the same membrane as the panel above it. (B) Fluorescent immuno-blot signals from panel A and from *Figure 6— figure supplement 1* were imaged and quantified using LI-COR Odyssey. The value of each methylation mark normalized to total H3 was calculated and the average of measurements from three independent clones, which serve as biological replicates, was plotted. The error bars represent standard error of the mean. Indicated p-values were generated by an unpaired, two-tailed (Student's) *t* test (*p≤0.01, **p≤0.005).

The following figure supplement is available for figure 6:

**Figure supplement 1.** Immuno-blot analysis of H3 histone methyl states from wild type (JRY10790) and *dld2Δ IDP2-R132H* mutants (JRY10734).

histone demethylase enzymes. The lack of change in H3K79 methylation levels was consistent with this model as there is no known H3K79 demethylase in *S. cerevisiae*.

## Enhanced gene silencing resulted from specific inhibition of two H3K36me demethylases

There are five known Jumonji-domain proteins in *S. cerevisiae* and among them Rph1, Gis1, and Jhd1 demethylate various species of methylated H3K36 (*Tu et al., 2007*; *Kim and Buratowski, 2007*) and Jhd2 demethylates mono-, di-, and trimethylated H3K4 (*Seward et al., 2007*). Because both H3K4 and H3K36 methylation increased, inhibition of multiple histone demethylases could potentially contribute to the increase in gene silencing. To identify the relevant histone demethylase (s) whose inhibition led to enhanced gene silencing, all five Jumonji-domain genes were deleted individually and in combinations and the resulting mutants assessed for a potential role in heterochromatin stability. The strong prediction was that deletion of the relevant histone demethylase(s) would phenocopy the effect of increased D2-HG levels if D2-HG caused increased gene silencing by inhibiting a histone demethylase. Indeed, both *gis1Δ* and *rph1Δ* single mutants enhanced gene silencing compared to wild type (p<0.0001; Student's *t* test), and the *rph1Δ gis1Δ* double mutants resulted in even further enhancement compared to wild type (p<0.0001; Student's *t* test) (*Figure 7A,B*). The

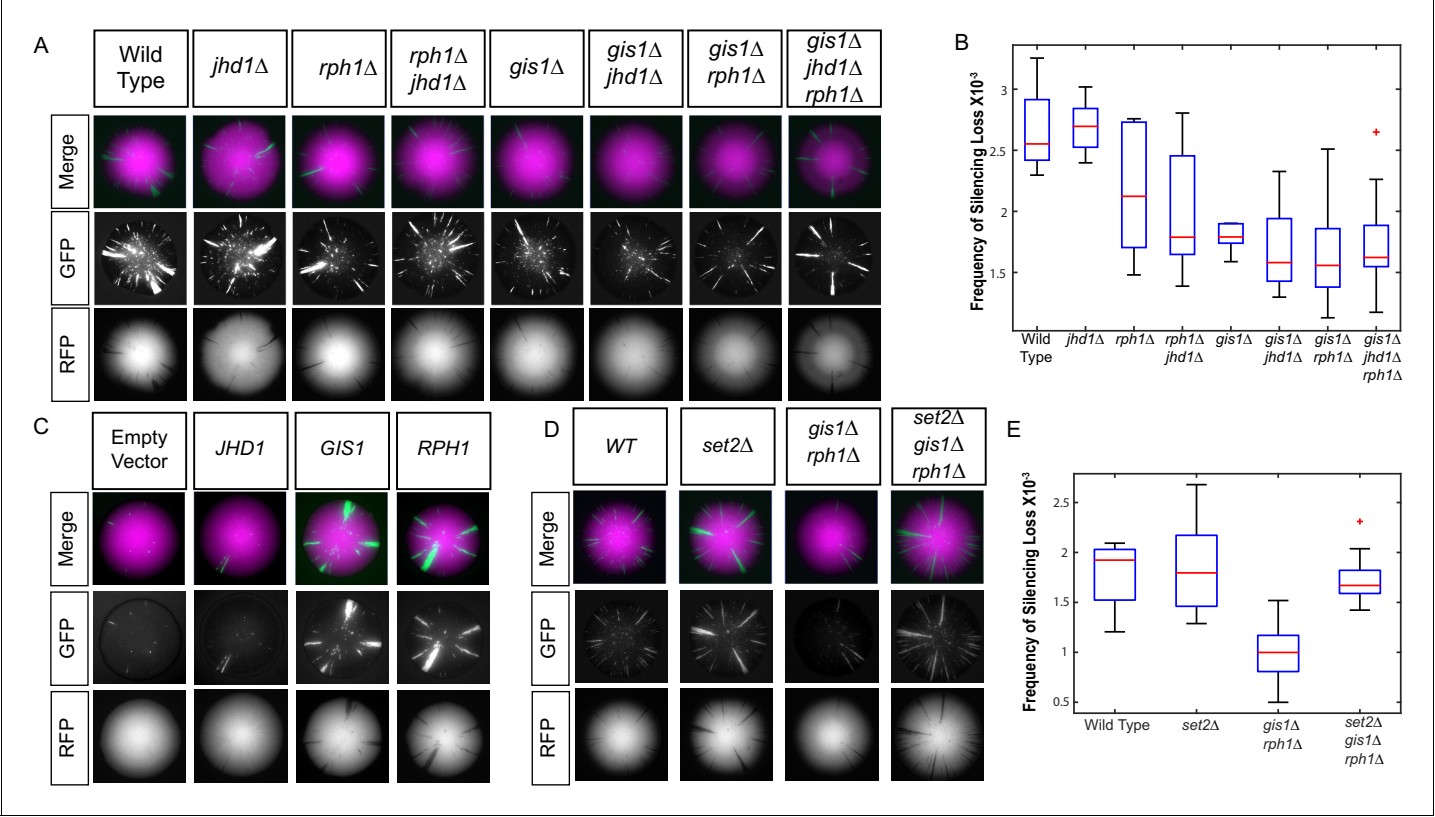

**Figure 7.** Loss of H3K36 demethylases Gis1 and Rph1 increased heterochromatin stability. (**A**) Images of CRASH assay reporter colonies that are wild type (JRY10790) or contain mutations in H3K36 demethylases. (**B**) Plots of the frequency of loss-of-silencing events from mutants in panel A calculated using MORPHE. (**C**) Images of CRASH assay reporter colonies (JRY10790) transformed with plasmids encoding Jumonji-domain demethylases overexpressed from a galactose-inducible promoter. The colonies were grown on agar plates containing 2% galactose. (**D**) Images of CRASH reporter strains with *set2Δ*, *gis1Δ*, and *rph1Δ* mutations. (**E**) Plots of the frequency of loss-of-silencing events from mutants in panel D calculated using MORPHE.

The following figure supplement is available for figure 7:

**Figure supplement 1.** Images of colonies with the CRASH reporter assay and *emc5Δ*, *jhd2Δ*, and *set1Δ* mutations grown on glycerol.

*jhd1Δ* mutant had no statistically significant effect on its own or in combination with either *rph1Δ* or *gis1Δ* mutations, nor did *ecm5Δ* and *jhd2Δ* mutations (*Figure 7—figure supplement 1*). Therefore, elevated H3K36 methylation was the determinant for increasing heterochromatin stability, and Rph1 and Gis1 were implicated as the primary demethylases inhibited by D2-HG accumulation that led to enhanced heterochromatic gene silencing. In further support, *GIS1* and *RPH1* overexpressed from a galactose-inducible plasmid led to destabilization of heterochromatin (*Figure 7C*); in contrast, over-expression of *JHD1* had no effect (*Figure 7C*).

In principle, the enhanced gene silencing in the *gis1Δ rph1Δ* double mutant could result from a failure to demethylate H3K36 methylated species, or from a failure to demethylate a here-to-fore unrecognized methylated species of any histone or non-histone protein. These possibilities were distinguished by the triple mutant combination *set2Δ gis1Δ rph1Δ*. Methylation of H3K36 is exclusively dependent on the methyltransferase Set2 (*Strahl et al., 2002*). The loss of Set2 completely abrogated the effect of *gis1Δ rph1Δ* on gene silencing. The extent of gene silencing in the triple mutant was no different from either wild type or *set2Δ* single mutants (*Figure 7D,E*). Because *set2Δ* suppressed *gis1Δ* and *rph1Δ*, the enhanced gene silencing in *gis1Δ* and *rph1Δ* mutants was due exclusively to H3K36 hypermethylation.

These results strongly predicted that *set2Δ* mutations would also suppress the phenotypes of *dld2Δ* and *IDP2-R132H* mutations. Indeed, gene silencing in *set2Δ dld2Δ* double mutants and *set2Δ dld2Δ IDP2-R132H* triple mutants was no different from *set2Δ* single mutants (*Figure 8A,B*). If D2-

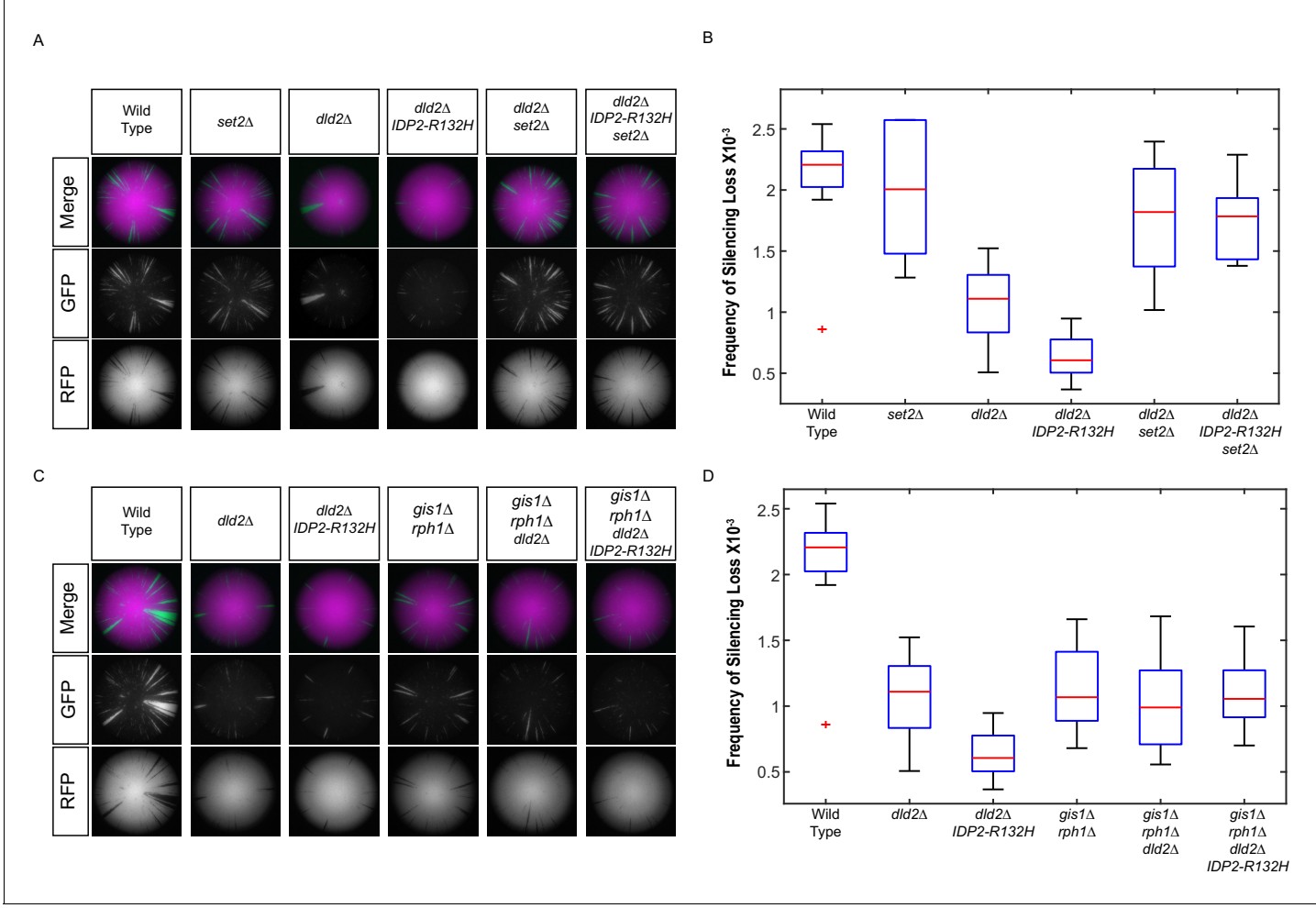

**Figure 8.** *IDP2-R132H* and *dld2Δ* silencing phenotypes depends on H3K36 hypermethylation. (**A**) Images of CRASH assay reporter colonies that are wild type (JRY10790) or mutant *dld2Δ* (JRY10733), *dld2Δ IDP2-R132H* (JRY10734), *set2Δ* (JRY10746), *dld2Δ set2Δ* (JRY10794), or *dld2Δ IDP2-R132H set2Δ* (JRY10795). (**B**) Plots of the frequency of loss-of-silencing events from mutants in panel A calculated using MORPHE. (**C**) Images of CRASH assay reporter colonies. The same strains as in panel A were used with the addition of *gis1Δ rph1Δ* (JRY10742), *gis1Δ rph1Δ dld2Δ* (JRY10792), and *gis1Δ rph1Δ dld2Δ IDP2-R132H* (JRY10793). (**D**) Plots of the frequency of loss-of-silencing events from mutants in panel C calculated using MORPHE. The experiments in panel A and panel C were performed together, and the same batches of colony images were used in MORPHE analysis for wild type, *dld2Δ*, and *dld2Δ IDP2-R132H* (panels B and D).

HG affects gene silencing by inhibition of H3K36 demethylases, then *gis1Δ rph1Δ* mutants should be epistatic to *dld2Δ* and *IDP2-R132H* mutations. This prediction was confirmed by analysis of *gis1Δ rph1Δ dld2Δ* and *gis1Δ rph1Δ dld2Δ IDP2-R132H* mutants, which displayed identical levels of silencing to the *gis1Δ rph1Δ* double mutant (*Figure 8C,D*). These results provided the functional confirmation that the elevated gene silencing observed in cells that accumulate D2-HG was due to Set2-dependent methylation of H3K36, and its subsequent hypermethylation due to inhibition of the Gis1 and Rph1 demethylases.

## Discussion

We have recapitulated cancer-associated IDH mutations in *S. cerevisiae* and demonstrated that the IDH mutants accumulated D2-HG in vivo and increased heterochromatin stability as reflected by enhanced gene silencing within heterochromatin. While DNA methylation is clearly impacted by D2-HG accumulation and leads to transcriptional changes in certain IDH-mutant tumors, it has been difficult to parse which cellular changes are attributable to inhibition of DNA demethylation, histone

demethylation, or other unexplored targets of demethylases. Here, we demonstrated unequivocally that enhanced gene silencing was due to histone H3K36 hypermethylation in response to the onco-metabolite D2-HG inhibiting two specific histone demethylases. As *S. cerevisiae* lacks DNA methylation, there were no confounding effects of DNA methylation in our study.

D2-HG has only been recently reported in *S. cerevisiae* (*Becker-Kettern et al., 2016*), and several new aspects of its metabolism are established here. D2-HG was detected in metabolite extracts even from wild-type cells, suggesting it is a normal component of the yeast metabolome. Wild-type cells maintained D2-HG at low levels through the activity of D2-HG dehydrogenases. Both Dld2 and Dld3 are capable of metabolizing D2-HG in vitro, and loss of either enzyme leads to accumulation of D2-HG in cells grown on glucose, with Dld3 having a stronger effect than Dld2 (*Becker-Kettern et al., 2016*). However, only Dld2 had a role in metabolizing D2-HG in vivo in cells grown on glycerol, suggesting that Dld3 and Dld2 are differentially regulated in response to the mode of energy production in the cell.

*IDP2-R132H* mutants alone, and in combination with *dld2Δ*, led to enhanced gene silencing. Multiple lines of evidence linked enhanced gene silencing to the production of D2-HG and inhibition of H3K36 histone demethylases Gis1 and Rph1. First, mutants that led to D2-HG accumulation phenocopied *gis1Δ rph1Δ* double mutants. Next, *gis1Δ rph1Δ* double mutants were epistatic to *IDP2-R132H* and *dld2Δ* mutations, indicating that the increase in gene silencing in *IDP2-R132H* and *dld2Δ* mutants was due to inhibition of these H3K36 demethylases. Additionally, the silencing phenotypes of H3K36 demethylase mutants, *IDP2-R132H,* and *dld2Δ* all depended on the H3K36 methyltransferase activity of Set2. Finally, external application of a cell-permeant form of D2-HG recapitulated enhanced silencing, demonstrating the causality of this phenotype. D2-HG acts as a competitive inhibitor of human Jumonji histone demethylase enzymes. Whether yeast Jumonji-domain histone demethylases are biochemically inhibited by D2-HG has not explicitly been tested, although the genetic and molecular evidence strongly implied such a mechanism.

D2-HG levels increased approximately 50-fold in *dld2Δ* mutants, and *IDP2-R132H* mutations also contribute to the pool of D2-HG, although this contribution was detectable only when a copy of wild-type *IDP2* was also present in *IDP2-R132H* mutant cells. This result was consistent with our genetic analysis of diploid cells in which silencing was greatest in *IDP2/IDP2-R132H* heterozygotes, and was also consistent with previous studies using human cell lines and tumors where the equivalent human mutation (*IDH1-R132H*) also partially depended on the presence an intact copy of wild-type *IDH1* for D2-HG production (*Jin et al., 2013*; *Ward et al., 2013*). Increases in D2-HG were readily measured in *dld2*-mutants, but not in cells capable of degrading D2-HG. This differs from tumor cells where IDH mutations lead to measurable increases in D2-HG despite intact D2-HG dehydrogenase activity (*Krell et al., 2011*). This discrepancy could either be due to a higher capacity of *S. cerevisiae* to metabolize D2-HG, a higher capacity for the human equivalent to the *IDP2-R132H* mutation to generate D2-HG, or some combination of both. We were able to detect a silencing phenotype in *IDP2-R132H* mutants even in cells capable of metabolizing D2-HG due to the highly sensitive nature of the CRASH assay, which converts transient lapses of silencing in individual cells into permanent and heritable effects. In contrast, physical measurements of metabolites reveal only steady-state levels averaged over millions of cells. Given that D2-HG is actively degraded when Dld2 is present, it is evident that our genetic assay was able to detect change caused by changes in D2-HG that were below our present level of physical detection.

Accumulation of D2-HG led to hypermethylation of histone H3 at lysine 4 and lysine 36. Importantly, it was the loss of histone demethylases that act on H3K36 that was solely responsible for increased stabilization of heterochromatin. Inhibition of H3K4 demethylation and the resulting elevation of bulk H3K4me had no effect on gene silencing. A role of H3K36 methylation in heterochromatin stability was surprising as other described roles for this modification involve co-transcriptional methylation of H3K36 to enhance transcriptional elongation (*Kizer et al., 2005*; *Tu et al., 2007*). Methylation of H3K36 at transcribed genes prevents aberrant transcription from initiating from within the open-reading frames of actively transcribed genes (*Carrozza et al., 2005*; *Keogh et al., 2005*; *Joshi and Struhl, 2005*), although this process appears to be incompatible with silencing at *HML* for reasons detailed below. Nevertheless, other aspects of histone methylation have had unexpected impacts on heterochromatic gene silencing. For example, H3K4 methylation occurs at the 5' ends of actively transcribed genes and is important for maintaining a local chromatin landscape conducive to transcription, and yet it also plays a role in the establishment of silencing at

*HML* (**Osborne et al., 2009**). Whether methylation of H3K4 is directly involved in the establishment and H3K36 in maintenance of *SIR*-dependent silencing remain important unresolved questions. Precedence for a direct role of histone methylation in silencing is evident in other species in which methylation of H3K9 and H3K27 directly recruit proteins that establish heterochromatin. Although budding yeasts lack the enzymes necessary to methylate H3K9 and K27, methylation of other histone lysines directly impact silencing. The site at which Sir3 interacts with histone H3 includes H3K79, and methylation of H3K79 by Dot1 inhibits Sir3-H3 interaction (**Armache et al., 2011**). The majority of euchromatic nucleosomes are methylated at H3K79, thus preventing Sir protein binding, while at heterochromatic loci Sir3 binds to H3 and sterically occludes Dot1 from accessing H3K79 for methylation. The results of this study are compatible with a model in which H3K36 methylation directly enhances the maintenance of silent chromatin, perhaps by recruitment of a pro-silencing factor to silent loci.

Alternatively, global increases in H3K36 methylation could recruit factors antagonistic to silencing away from heterochromatin. For example, the Rpd3 complex antagonizes silencing at heterochromatic loci *HML* and *HMR* by limiting Sir3 and Sir4 occupancy at flanking silencer regions (**Thurtle-Schmidt et al., 2016**). Importantly, the deacetylase activity of Rpd3(S) is directed toward nucleosomes at actively transcribed genes in part through an interaction between the Eaf3 subunit of the Rpd3(S) complex and histones methylated at H3K36 (**Carrozza et al., 2005**; **Keogh, 2005**; **Joshi and Struhl, 2005**). It is possible that elevated H3K36 methylation in euchromatin recruits Rpd3(S) away from *HML* and *HMR,* and partially alleviates its antagonistic effect on silencing. Because recruitment of Rpd3 to heterochromatin appears to be independent of Eaf3 and thus H3K36 methylation (**Thurtle-Schmidt et al., 2016**), a global increase of H3K36 methylation could lead to relocalization of a limited pool of Rpd3 away from *HML* and *HMR* to sites with increased H3K36 methylation. Finally, H3K36 methylation prevents binding of Sir proteins to heterochromatin-adjacent and telomere-adjacent regions (**Tompa and Madhani, 2007**). Increased global H3K36 methylation due to D2-HG accumulation could, in principle, prevent binding of Sir proteins to non-heterochromatic regions, making more Sir protein available to bind at heterochromatic loci.

Increased silencing of tumor suppressor genes is a feature of many tumors. This study pinpointed specific histone demethylase enzymes whose inhibition by D2-HG resulted in increased gene silencing. Further, this study demonstrated that D2-HG enhanced gene silencing through stabilization of heterochromatic DNA. Similar mechanisms could promote tumor formation by silencing genes encoding tumor suppressors.

## Materials and methods

### Yeast strains and plasmids

The strains used in this study are derived from the W303 background and listed in **Supplementary file 1**. Components of the CRASH assay were generated as described previously (**Dodson and Rine, 2015**). Gene deletions were made by one-step integration of disruption cassettes (**Goldstein and McCusker, 1999**; **Goldstein et al., 1999**; **Wach et al., 1994**) and confirmed by PCR. The *IDP2-R132H* point mutation was generated by adaptamer-mediated PCR (**Erdeniz et al., 1997**). A PCR fragment of *IDP2* was amplified using genomic DNA template and IDP2-R132H Ad-A Mut Forward primer with IDP2-835-814 Ad-B Reverse primer (Fragment 1). Truncated, overlapping fragments of *URA3* $^{K.\ lactis}$ were amplified using Adaptamer b URA3kl Forward with *K. lactis* 3' internal reverse primer (Fragment 2) and *K. lactis* 5' internal forward primer with Adaptamer a URA3kl reverse primer (Fragment 3). PCR fragments 1 and 2 were mixed in a PCR reaction and fused together using primers IDP2-R132H MutA Forward and *K. lactis* 3' internal reverse, resulting in Fragment L. PCR fragments 1 and 3 were mixed in a PCR reaction and fused together using primers *K. lactis* 5' internal forward and IDP2-835-814 Ad-B Reverse, resulting in Fragment R. Allele replacement of *IDP2* was performed by co-transforming fragments L and R into JRY10790 and selection on CSM-URA to generate a direct repeat of *IDP2-R132H* separated by *URA3*$^{K.lactis}$. The resulting transformants were streaked on 5-FOA to select for recombination events resulting in a single copy of *IDP2-R132H* at the native *IDP2* locus and loss of the second copy of *IPD2-R132H* and *URA3*$^{K.lactis}$. The presence of a single copy of *IPD2-R132H* was confirmed by PCR and sequencing of the entire ORF. The *CEN-ARS-LEU2* plasmid (pRS315) (**Sikorski and Hieter,**

*1989*) was used to generate pJR3399. A PCR fragment containing the *IDP2* open reading frame and 319 bases of the *IDP2* 5'UTR and 292 bases of the *IDP2* 3'UTR was amplified from *S. cerevisiae* genomic DNA template using the Gibson Assembly-compatible forward primer 5'-AATTAACCC TCACTAAAGGCTCCGGGATGTCATTGCCGG-3' and reverse primer 5'-TAATACGACTCACTA TAGGGTCCTCTGTGTAGGTTGTAACGA-3' and was inserted into the multiple cloning site of pRS315 by Gibson Assembly (New England Biolabs, Ipswich, MA). In *Figure 5*, the indicated strains were transformed with either an empty vector *CEN-ARS-HIS3* plasmid (pRS413) (*Sikorski and Hieter, 1989*) or a plasmid with *cre* expressed from the *GAL1* promoter (pSH62) (*Gueldener et al., 2002*).

## Yeast metabolite extractions

Cells were grown to mid-log phase in minimal medium with glycerol as the sole carbon source. Prior to harvesting, $1 \times 10^8$ cells were mixed with 40 ml of 60% methanol chilled to −20°C. Cells were pelleted by centrifugation at 4°C at 1540 x g for 5 min and pellets were frozen in liquid nitrogen. Pellets were resuspended in 2.5 ml of 100% methanol (−20°C), refrozen in liquid nitrogen, and immediately thawed in an ice bath. The suspension was pelleted by centrifugation at 770 x g for 20 min at 4°C. The resulting supernatant was collected and stored at −20°C. An additional round of extraction was performed by resuspending the pellet in 2.5 ml of methanol and vortexing for 30 s at 4°C. Insoluble material was pelleted by centrifugation as before and the resulting supernatant was pooled with the first extraction. A 0.5 ml aliquot of the extract was transferred to a new tube and the methanol was evaporated by vacuum concentrator (Savant SpeedVac, Thermo Scientific, Waltham, MA) for 1 hr without heating. The resulting residue was resuspended in 100 μl of water immediately prior to liquid chromatography-mass spectrometry.

## Liquid chromatography-mass spectrometry

Metabolite extract samples were analyzed using an Agilent 1200 liquid chromatograph (LC; Santa Clara, CA) that was connected to an LTQ-Orbitrap-XL mass spectrometer equipped with an electrospray ionization (ESI) source (Thermo Fisher Scientific, Waltham, MA). The LC was equipped with an in-line filter (KrudKatcher Classic, Phenomenex, Torrance, CA), a C18 analytical column (Atlantis T3, 150 mm length ×1.0 mm inner diameter, 3 μm particles, Waters, Milford, MA) and a 100-μL sample loop. Acetonitrile, formic acid, and methanol (Fisher Optima grade, 99.9%), and water purified to a resistivity of 18.2 MΩ·cm (at 25°C) using a Milli-Q Gradient ultrapure water purification system (Millipore, Billerica, MA) were used to prepare mobile phase solvents. Solvent A was 99.9% water/0.1% formic acid and solvent B was 50% acetonitrile/50% methanol (v/v). The elution program consisted of isocratic conditions at 1% B for 2 min, a linear gradient to 85% B over 2 min, isocratic conditions at 85% B for 2 min, a linear gradient to 1% B over 1 min, and isocratic conditions at 1% B for 20 min, at a flow rate of 50 μL/min. The column and sample compartments were maintained at 30°C and 4°C, respectively. The column exit was connected to the ESI probe of the mass spectrometer using PEEK tubing (0.005' inner diameter ×1/16'' outer diameter, Agilent). Mass spectra were acquired in the negative ion mode over the range *m/z* = 100 to 1000 using the Orbitrap mass analyzer, in profile format, with a resolution setting of 100,000 (measured at *m/z* = 400, full width at half-maximum peak height). Mass spectrometry data acquisition and analysis were performed using Xcalibur software (version 2.0.7, Thermo) and quantitative analysis calculations were performed using Excel (Office Professional Plus 2013, Microsoft, Redmond, WA). Averages for each strain were determined from three independent clones which serve as biological replicates and divided by the average calculated for wild type. Unpaired two-sided (Student's) *t* tests were used to determine whether differences in metabolite levels were statistically significant.

## Colony growth and imaging

Cells were diluted and plated onto 1.5% agar plates containing complete supplement mixture (CSM) – Trp (Sunrise Science Products, San Diego, CA), CSM – Trp – His (Sunrise Science Products, San Diego, CA), or, as indicated, yeast nitrogen base (YNB) without amino acids (Difco-Beckton Dickinson, Franklin Lakes, NJ) and either 3% glycerol, 2% glucose, 2% raffinose, or 2% galactose as indicated. Strains transformed with plasmids pRS315 and pJR3399 were grown using media exactly as described above but with the dropout mixture CSM – Trp – Leu (Sunrise Science Products, San

Diego, CA). Strains transformed with plasmids pRS413 and pSH62 were grown exactly as described above but with the dropout mixture CSM – Trp – His (Sunrise Science Products, San Diego, CA). Colonies were grown for 10 days at 30°C unless otherwise noted. All colonies were imaged using a Zeiss Axio Zoom.V16 microscope equipped with ZEN software (Zeiss, Jena, Germany), a Zeiss AxioCam MRm camera and a PlanApo Z 0.5× objective. The tops of colonies were imaged in RFP and GFP channels. The RFP and GFP channels are shown separately and merged for a representative colony image for each strain.

## Immuno-blotting

Cells were grown in liquid cultures of CSM-Trp-3% glycerol shaking at 30°C until mid-log phase. 10 OD units of cells were harvested from each culture, pelleted, frozen with liquid nitrogen, and stored at −80°C. Pellets were resuspended in 200 µl of 20% w/v trichloroacetic acid and transferred to 2-ml screw cap tubes. Cell extracts were prepared by addition of an equal volume of 0.5 mm zirconium ceramic beads (BioSpec Products, Bartlesville, OK) followed by bead beating using a Millipore MP-20 FastPrep (EMD Millipore, Billerica, MA) on setting 5.5 with 20 s cycle duration. A total of five cycles were performed with 2-min incubations on ice between each run to prevent overheating of sample. Recovered precipitate was dissolved in 200 µl of 2X Laemmli buffer and the pH adjusted by adding 30 µl of 1.5 M Tris pH = 8.8. Samples were heated at 65°C for 10 min and insoluble material was pelleted by centrifugation. An equal amount of the soluble portion of each sample was run on SDS-polyacrylamide gels and transferred to nitrocellulose membranes. Membranes were blocked in Li-Cor Odyssey Blocking Buffer (LI-CORE Biosciences, Lincoln, NE) and the following primary antibodies were used for immunodetection: anti-histone H3 mono methyl K36 (Ab9048), anti-histone H3 di methyl K36 (Ab9049), anti-histone H3 tri methyl K36 (Ab9050), anti-histone H3 tri methyl K4 (Ab12209), and $\alpha$-H3 (Abcam Ab1791) were from Abcam, Cambridge, U.K., anti-histone H3 tri methyl K79 (pAB-068–050) from Diagenode, Denville, NJ, and anti-phosphoglycerate kinase antibody (22C5D8) from ThermoFisher, Rockford, IL. The specificity of each histone methyl antibody was confirmed by competition experiments with purified peptides. Membranes were incubated with infrared dye-conjugated secondary antibodies, IRDye800CW goat anti-mouse and IRDye680RD goat anti-rabbit antibodies (LI-CORE Biosciences, Lincoln, NE) and imaged on a LI-CORE Odyssey imager in the 700 nm and 800 nm channels. All washing steps were performed with Tris-buffered saline – 0.05% Tween-20. Quantitative analysis of immunoblots was performed using LI-CORE Image Studio software (LI-CORE Biosciences, Lincoln, NE). Unpaired two-sided (Student's) $t$ tests were used to determine whether differences in histone methylation were statistically significant.

## MORPHE quantification of loss of silencing events

Quantification of loss of silencing events was performed using MORPHE software as described (*Liu et al., 2016*). To avoid possible effects of batch-to-batch variation, quantitative comparisons between strains were limited to experimental batches defined as those plated on the same day using the same batch of media. The batches were grown and imaged as a group on the same days. Microscope settings and software parameters were kept uniform within batches. For plotting, the average frequency of silencing loss within individual colonies was calculated. All the values obtained for each colony were then used to generate box plots. The red line represents the median value calculated from at least eight colonies. The blue boxes represent the 25th and 75th percentile. Whiskers represent the range of values within 1.5-times the interquartile range. Values extending beyond 1.5-times the interquartile range are marked as outliers (Red +). Unpaired two-sided (Student's) $t$ tests were used to determine whether differences in frequency of silencing loss were statistically significant.

## Acknowledgements

We are grateful to Michael Botchan, Michael Marletta and Barbara Meyer for valuable feedback on the manuscript. This work was supported by grants from the National Institutes of Health (GM31105 and GM120374 to JR, 1S10OD020062-01 to ATI, and F32 GM115074 to RJ).

## Additional information

### Funding

| Funder | Grant reference number | Author |
|---|---|---|
| National Institutes of Health | F32 Postdoctoral Fellowship, GM115074 | Ryan Janke |
| National Institutes of Health | 1S10OD020062-01 | Anthony Iavarone |
| National Institutes of Health | GM31105 | Jasper Rine |
| National Institutes of Health | GM120374 | Jasper Rine |

The funders had no role in study design, data collection and interpretation, or the decision to submit the work for publication.

### Author contributions

RJ, Conceptualization, Data curation, Formal analysis, Funding acquisition, Investigation, Writing—original draft, Writing—review and editing; ATI, Data curation, Methodology, Writing—original draft; JR, Conceptualization, Supervision, Funding acquisition, Writing—original draft, Writing—review and editing

### Author ORCIDs

Ryan Janke, http://orcid.org/0000-0002-5755-8019
Jasper Rine, http://orcid.org/0000-0003-2297-9814

## Additional files

### Supplementary files

• Supplementary file 1. *Saccharomyces cerevisiae* strains used in this study. All strains were derived from W303 with the following genotype: *can1-100 his3-11 leu2-3,11 lys2 TRP1 ADE2 ura3-1.*

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
