## [Decision Letter]

Thank you for submitting your article "Oncometabolite D-2-Hydroxyglutarate enhances gene silencing through inhibition of specific H3K36 histone demethylases" for consideration by *eLife*. Your article has been reviewed by three peer reviewers, one of whom, Alan Hinnebusch (Reviewer #1), is a member of our Board of Reviewing Editors, and the evaluation has been overseen by Kevin Struhl as the Senior Editor.

The reviewers have discussed the reviews with one another and the Reviewing Editor has drafted this decision to help you prepare a revised submission.

Summary:

This study shows that The IDP2-R132H mutation in isocitrate dehydrogenase (IDH; encoded by IDP2 in yeast) provokes formation of the metabolite 2-HG, associated with tumor formation or progression in humans. The results show that IDP2-R132H enhances silencing within the HML locus, whereas deleting IDP2 does not; and that eliminating the gene product of DLD2, one of two possible yeast orthologs of the human enzyme D2HGDH that metabolizes 2-HG, also reduces silencing, even in cells containing WT IDP2 but in a manner exacerbated by the IDP2-R132H mutation, thus indicating that 2-HG production in yeast doesn't require, but is enhanced by the IDP2 tumor-associated mutation. This conclusion is supported by measuring 2-HG production in extracts, which is very high in the dld2 mutant, and somewhat higher in the double mutant also containing the IDP2-R132H mutation; but the latter mutation alone does not suffice to increase 2-HG levels by a detectable amount. They further show that addition of exogenous octyl-D2-HG increases HML silencing in the dld2 mutant background. Thus, they establish significant correlations between 2-HG levels and increased HML silencing in the dld2 mutant, which is defective for 2-HG metabolism. They go on to show that H3-K36 and H3K4 methylation is elevated in the dld2 IDP2-R132H double mutant, and that HML silencing is increased in two mutants lacking histone demethylases specific for methylated H3-K36, Rph1 and Gis1, but not in three other histone demethylase mutants; however, the specific involvement of these two demethylases is not explained at the level of increased H3-K36 methylation in the rph1 and gis1 mutants versus the other three demethylase mutants. They conclude by demonstrating that deleting RPH1 and GIS1 suppresses the effects on silencing associated with increased 2-HG production conferred by the dld2 mutation, in the manner expected if 2-HG enhances silencing by inhibiting these two demethylases. Finally, they show that eliminating Set2 suppresses the effect of elevated 2-HG in dld2 cells, as would be expected if elevated methylated H3-K36 is responsible for the increased silencing. Based on these results they conclude that 2-HG accumulation in cells enhances HML silencing by inhibiting two specific H3-K36 demethylases with attendant accumulation of methylated H3-K36.

Essential revisions:

It is necessary to extend the CRASH assay and demonstrate a difference in silencing between the two dld2 strains harboring WT IDP2 or IDP2-R132H that were analyzed in Figure 3. It is necessary quantify the immunoblot data in Figure 6 using normalization to H3. It is necessary to explain fully what statistical tests have been applied, supply error bars to Figure 3, and histograms summarizing the results in Figure 7—figure supplement 1.

The authors are advised to draw the distinction between a biochemical demonstration that 2-HG inhibits histone demethylases in vitro and genetic evidence supporting this possibility for the yeast demethylases Rph1 and Gis1. While not essential for acceptance, it was considered that the scientific quality of the work would be increased by comparing the effects of each of the demethylase mutants on bulk H3-K36 methylation by Western analysis in an effort to explain why only the RPH1 and GIS1 deletions affect HML silencing, or cite the appropriate published data; and also examining whether accumulation of D2-HG in human cells results in histone hypermethylation.

Reviewer #1:

This study shows that The IDP2-R132H mutation in isocitrate dehydrogenase (IDH; encoded by IDP2 in yeast) provokes formation of the metabolite 2-HG, associated with tumor formation or progression in humans. The results show that IDP2-R132H enhances silencing within the HML locus, whereas deleting IDP2 does not; and that eliminating the gene product of DLD2, one of two possible yeast orthologs of the human enzyme D2HGDH that metabolizes 2-HG, also reduces silencing, even in cells containing WT IDP2 but in a manner exacerbated by the IDP2-R132H mutation, thus indicating that 2-HG production in yeast doesn't require, but is enhanced by the IDP2 tumor-associated mutation. This conclusion is supported by measuring 2-HG production in extracts, which is very high in the dld2 mutant, and somewhat higher in the double mutant also containing the IDP2-R132H mutation; but the latter mutation alone does not suffice to increase 2-HG levels by a detectable amount. They further show that addition of exogenous octyl-D2-HG increases HML silencing in the dld2 mutant background. Thus, they establish significant correlations between 2-HG levels and increased HML silencing in the dld2 mutant, which is defective for 2-HG metabolism. They go on to show that H3-K36 and H3K4 methylation is elevated in the dld2 IDP2-R132H double mutant, and that HML silencing is increased in two mutants lacking histone demethylases specific for methylated H3-K36, Rph1 and Gis1, but not in three other histone demethylase mutants; however, the specific involvement of these two demethylases is not explained at the level of increased H3-K36 methylation in the rph1 and gis1 mutants versus the other three demethylase mutants. They conclude by demonstrating that deleting RPH1 and GIS1 suppresses the effects on silencing associated with increased 2-HG production conferred by the dld2 mutation, in the manner expected if 2-HG enhances silencing by inhibiting these two demethylases. Finally, they show that eliminating Set2 suppresses the effect of elevated 2-HG in dld2 cells, as would be expected if elevated methylated H3-K36 is responsible for the increased silencing. Based on these results they conclude that 2-HG accumulation in cells enhances HML silencing by inhibiting two specific H3-K36 demethylases with attendant accumulation of methylated H3-K36.

General critique:

Overall, the results justify the conclusions of the paper, which are significant in providing strong evidence that inhibition of certain H3-K36 demethylases is a consequence of the accumulation of 2-HG in cells. This is important because it is currently unclear whether inhibition of these enzymes, or rather of DNA methylases (lacking in yeast) is involved in the oncogenic properties of this metabolite in human cells. Others have shown that 2-HG can inhibit mammalian Jumonji histone demethylases in vitro. While this study does not establish this point directly for the yeast demethylases Rph1 and Gis1, it provides compelling genetic evidence that they are likely inhibited by 2-HG in vivo. It seems advisable that the authors draw this distinction between biochemical demonstration and strong inference from genetic data. Apart from this, the authors need to provide better documentation of the statistical tests that were conducted for certain experiments, as detailed below. In addition, it seems important that they analyze the effects of each of the five demethylase mutants on bulk H3-K36 methylation by Western analysis to provide insights into why only the RPH1 and GIS1 deletions affect HML silencing. Is it the case that only these two deletions provoke a strong increase in methylated H3-K36 species, or is something more complicated involved, such as locus-specific effects of the enzymes with only RPH1 or GIS1 functioning at HML (see specific comment below for more on this request.)

Major comments:

– Subsection “Mutations expected to produce D2-HG results in enhanced gene silencing”: the statistical test employed and the specific comparisons to which the p values were calculated need to be indicated in text, figure or legend. (This also applies to the statements about Figure 7 in subsection “Enhanced gene silencing resulted from specific inhibition of two H3K36me 211 demethylases”) It would be easy to indicate this in the figure with a bracket connecting the data being compared with the p-value, as done in Figure 3. Notched box plots could also be used instead in Figure 1–Figure 2 and Figure 7.

– Figure 3: error bars need to be added to these histograms.

– Figure 7—figure supplement 1: histograms summarizing the results of replicate measurements and statistical analysis should be presented here, as in other similar figures.

– As only the gis1 and rph1 mutants enhanced silencing, and the other three demethylase mutants did not, the question arises about whether the gis1 and rph1 mutations have a relatively greater effect on bulk H3-K36 demethylation. This is to be expected for Jhd2, which was cited as being specific for methylated H3-K4, but is apparently unknown for Ecm5, and the relative efficacies of Gis1, Rph1, and Jhd1 for demethylating H3-K36 species in vivo was not addressed. Unless it is known from the literature, the effects of each demethylase mutant on bulk H3-K36 methylated species should be tested using the Western assay in Figure 6, and the results discussed appropriately. Observing some change in methylation in the other 3 mutants would also provide confidence that the strains not only lack the relevant demethylase gene product but also have not acquired a suppressor mutation that increases the expression/activity of one of the other demethylases still present. In fact, paragraph five of the Discussion section implies that this experiment has already been done for the jhd2 mutant.

– Discussion section, paragraph five: More pertinent to silencing is the known role of H3-K36 methylation by Set2 in suppressing cryptic promoters in coding sequences.

Reviewer #2:

In this paper, the authors explore the relationship between the accumulation of D-2-hydroxyglutarate (D-2 HG) and gene silencing, emphasizing the role of D-2 HG in inhibition of several H3K36 histone demethylases. As levels of the metabolite D-2 HG are normally low in human cells but become elevated in tumor cells (in which isocitrate dehydrogenases are mutated), the authors focus on the impact of loss or mutation of the *S. cerevisiae* ortholog IDP2, as well as loss of DLD2, a D-2 HG dehydrogenase. Using the CRASH assay to detect changes in gene expression at a reporter locus, the authors focus on inhibition of histone demethylation without the confounding influence of DNA methylation/demethylation. The authors conclude that the demethylases Rph1 and Gis1 are inhibited by D2-HG, leading to increased H3K36 methylation and gene repression at this locus.

While these results are of interest, they do not appear to unveil new biological insights regarding histone demethylation and/or silencing to merit inclusion in *eLife*. Unfortunately the authors do not revisit the IDP2 mutation in human tissue culture and thus, in its current form, this work does not significantly advance our understanding of histone demethylation, D-2 HG regulation in yeast, gene silencing or how this mechanism functions to inhibit histone demethylation in the context of human disease. While the work itself is interesting, these studies do not establish a direct link between inhibition of demethylation and repression of gene silencing, leaving open the possibility of an indirect mechanism. The authors may want to consider addressing additional key questions such as when in the yeast life cycle or under what stress conditions might up-regulation of D-2 HG result in a meaningful biological outcome with respect to histone methylation or gene silencing.

In addition, there are several major and minor concerns to address:

1) One significant problem is that the D-2 HG levels do not increase for the IDP2-R132H mutant when compared to wild type using LC-MS in Figure 3, contrasting with the CRASH results in Figure 1. The authors attempt to address this apparent discrepancy by transforming with a plasmid with native promoter-driven IDP2 (Figure 3), but they do not then revisit the CRASH assay with plasmid-based IDP2 overexpression. Furthermore, Figure 3 lacks error bars. Since the Materials and methods section does not describe any plasmids used in the paper, plasmid copy number is not addressed.

2) The immunoblot quantifications in Figure 6 should be normalized to H3 in some manner – preferably using LI-COR to probe for H3 methyl marks and H3 on the same membrane.

3) The authors state that there is no H3K79 demethylase in *S. cerevisiae*, which is a much stronger statement than that none has been identified.

4) The 2007 Genetics paper by Tompa & Madhani should be referenced.

5) All instances of the term "Li-Core" should reflect the true name "LI-COR."

Reviewer #3:

Rine and colleagues use elegant yeast genetics to uncover a novel connection between metabolism and chromatin regulation through the use of specific cancer mutations and in the process gain a deeper understanding about cancer biology. Specifically, the authors introduce into the model organism *S. cerevisiae* mutations in the IDH homolog IDP2, which is highly conserved, and measure their effects on gene silencing. They observed a drastic increased stability on gene silencing though stabilization of heterochromatin, an effect that was dominant when compared to WT. Mutants (dld2 and dld3) that were predicted to increase levels of D2-HG also mediated the same effect. These metabolic effects were confirmed using MS and an enhanced effect was observed when the dld2 mutant was combined with the IDH mutation. Finally, exogenous D2-HG was added and in dld2 mutants, significant stabilization of gene silencing was observed. Collectively this provides strong evidence that there is indeed a functional connection between D2-HG levels and chromatin regulation. Further mechanistic insight is revealed as increase of H3K36 and H3K4 (but not K79) methylation is observed in the mutant backgrounds and a strong connection is uncovered through the jumonji demethylases.

The paper is well written and beautifully demonstrates the value of pivoting back and forth between human genomic datasets and work in more genetically tractable systems to uncover important mechanistic understanding behind cancer biology. There are several major novel themes in the paper including: 1) connecting key IDH cancer mutations to chromatin silencing through histone methylation/demethylation; 2) linking more tightly metabolism and chromatin function; 3) uncovering rare, interesting phenotypes for jumoji demethylases, and 4) demonstration of the power of yeast genetics to functionally characterize cancer mutations. It is because of this collection of novelty I strongly feel that this paper will be of great interest to the wide readership of *eLife*, and therefore I support publication.

---

## [Author Response]

*Reviewer #1:*

[…]

*General critique:*

*[…] Others have shown that 2-HG can inhibit mammalian Jumonji histone demethylases* in vitro*. While this study does not establish this point directly for the yeast demethylases Rph1 and Gis1, it provides compelling genetic evidence that they are likely inhibited by 2-HG* in vivo*. It seems advisable that the authors draw this distinction between biochemical demonstration and strong inference from genetic data. […]*

Previous detailed in vitro biochemical studies demonstrate that D2-HG acts as a competitive inhibitor of mammalian Jumonji histone demethylases. As the reviewer carefully points out, we did not directly test the biochemical impact of D2-HG on yeast Jumonji demethylases, however we have provided compelling genetic evidence to support such a model. It was suggested by the reviewer that we should clearly draw the distinction between our experiments and direct biochemical demonstration of competitive inhibition. At the recommendation of the reviewer we have added wording to distinguish this point in the Discussion section. We thank the reviewer for their careful attention to detail.

*Major comments:*

*– Subsection “Mutations expected to produce D2-HG results in enhanced gene silencing”: the statistical test employed and the specific comparisons to which the p values were calculated need to be indicated in text, figure or legend. (This also applies to the statements about Figure 7 in subsection “Enhanced gene silencing resulted from specific inhibition of two H3K36me 211 demethylases”) It would be easy to indicate this in the figure with a bracket connecting the data being compared with the p-value, as done in Figure 3. Notched box plots could also be used instead in Figure 1–Figure 2 and Figure 7.*

The statistical tests employed in Figure 1–Figure 2 and Figure 7 have now been clearly defined in the text. The specific samples being compared in each statistical analysis are also now clearly defined in the text.

*– Figure 3: error bars need to be added to these histograms.*

Error bars representing standard error of the mean have been added to Figure 3 and additional statistical testing has been employed. Samples that were significantly different (p < 0.05) from wild type (student’s t test) have been annotated with an asterisk (*) placed above the bar corresponding to that sample within the figure. A description of the statistical analysis and the annotation used in the figure has been added to Figure 3’s legend.

*– Figure 7—figure supplement 1: histograms summarizing the results of replicate measurements and statistical analysis should be presented here, as in other similar figures.*

Figure 7—figure supplement 1 data have been quantified using MORPHE software and the resulting plots have been added to the figure. Statistical tests used to compare the samples have now been defined in the figure legend. As expected, no statistically significant differences were observed between wild-type and mutant samples tested in Figure 7—figure supplement 1. These results are reported in the text (subsection “Enhanced gene silencing resulted from specific inhibition of two H3K36me demethylases.”) as well as the figure legend.

*– As only the gis1 and rph1 mutants enhanced silencing, and the other three demethylase mutants did not, the question arises about whether the gis1 and rph1 mutations have a relatively greater effect on bulk H3-K36 demethylation. This is to be expected for Jhd2, which was cited as being specific for methylated H3-K4, but is apparently unknown for Ecm5, and the relative efficacies of Gis1, Rph1, and Jhd1 for demethylating H3-K36 species* in vivo *was not addressed. Unless it is known from the literature, the effects of each demethylase mutant on bulk H3-K36 methylated species should be tested using the Western assay in Figure 6, and the results discussed appropriately. Observing some change in methylation in the other 3 mutants would also provide confidence that the strains not only lack the relevant demethylase gene product but also have not acquired a suppressor mutation that increases the expression/activity of one of the other demethylases still present. In fact, paragraph five of the Discussion section implies that this experiment has already been done for the jhd2 mutant.*

The reviewer brings up a counter hypothesis (thank you) to understand why the H3K36 demethylases Gis1 and Rph1, but not the other H3K36 histone demethylase Jhd1, are the critical targets underlying the effect of D2-HG on gene silencing. The counter hypothesis cleverly pointed out by the reviewer, is that Gis1 and Rph1 might have an effect, not from their specificities, but rather from having a greater effect on bulk H3-K36 demethylation compared to the other H3K36 demethylase Jhd1. The reviewer suggests a simple test of this idea by mutating each H3K36 demethylase (and the potential effect of Ecm5) and measuring the effect on bulk H3K36 methylation.

As the reviewer inferred, such an experiment has been conducted and the effect of deleting each individual H3K36 demethylase has been published. No effect on the bulk methylation levels of histones are observed by immunoblot in any single histone demethylase mutant (Kwon and Ahn (2011) Role of yeast JmjC-domain containing histone demethylases in actively transcribed regions. Biochem. Biophys. Res. Commun., 410(3): p614-619). This demonstrates that the specificity we observe for Gis1 and Rph1 is not simply explained by those demethylases individually having a greater general effect on the bulk levels of H3K36 methylation compared to Jhd1 or other demethylases. Another study on locus-specific effects (rather than bulk effects) of H3K36 demethylase mutants in yeast demonstrates that H3K36 demethylation occurs at the same rate in single H3K36 demethylase mutants (*rph1Δ, gis1Δ*, or *jhd1Δ*) as it does in wild-type cells, while an *rph1Δ gis1Δ jhd1Δ* triple mutant has delayed turnover of H3K36 methylation (Sein et al. (2015) Distribution and maintenance of histone H3 lysine 36 trimethylation in a transcribed locus. PLoS One, 10(3)).

Collectively, these data suggest that analysis of H3K36 methylation levels for bulk H3 or at specific loci in single demethylase mutants will likely not reveal any insight into the relative contribution of each demethylase beyond the studies that have already been performed.

The reviewer points out that it is possible that the lack of a phenotype in a *jhd1Δ* mutant could be explained by the presence of a suppressor mutation that increases expression/activity of the other H3K36 demethylases. In this scenario, the contribution of *jhd1Δ* would then be expected to be revealed in the *rph1Δ gis1Δ jhd1Δ* triple mutant, where the effect of the potential suppressor mutation would be lost. We did not observe any additional contribution of *jhd1Δ* to the silencing phenotype in this triple mutant (Figure 7) nor was there a growth deficiency that would have been a selection for suppressor mutations.

*– Discussion section, paragraph five: More pertinent to silencing is the known role of H3-K36 methylation by Set2 in suppressing cryptic promoters in coding sequences.*

We demonstrated that increased H3K36 methylation increased silencing within heterochromatin. In our discussion we stated that the role of this mark on silent heterochromatin was surprising because it is generally found at actively transcribed loci and thought to promote a chromatin architecture conducive to active transcription. The reviewer importantly points out that a discussion regarding the known role of H3K36 methylation in suppressing cryptic promoters in coding sequences is particularly warranted in the context of our study, and we completely agree. At the suggestion of the reviewer, a discussion on this point was added to the Discussion section.

In short, the reviewer’s suggestion is that at actively transcribed sites, H3K36 methylation recruits the RPD3 histone deacetylase complex, which removes acetyl marks on histone H4, and that this recruitment is playing a role our observations. It has been demonstrated that this deacetylation is necessary to suppress transcription from cryptic promoters within coding sequences. This lends to the model that perhaps hypermethylation of H3K36 within heterochromatin could increase recruitment of RPD3 to heterochromatin and subsequent deacetylation could increase silencing.

However, such a model is not compatible with the fact that recruitment of RPD3 to silent heterochromatin is actually antagonistic to silencing (Thurtle-Schmidt, Dodson, and Rine, 2015. In light of this, we also discuss a different model that is more compatible with the nuanced regulation of silent heterochromatin. This model is centered on the possibility that the increase in bulk H3K36 methylation could recruit a limited pool of RPD3 away from heterochromatin and thus reduce the antagonistic effect of RPD3 on silencing that has been previously characterized. The additional discussion inspired by the reviewer’s comment has strengthened the manuscript.

*Reviewer #2:*

*In this paper, the authors explore the relationship between the accumulation of D-2-hydroxyglutarate (D-2 HG) and gene silencing, emphasizing the role of D-2 HG in inhibition of several H3K36 histone demethylases. As levels of the metabolite D-2 HG are normally low in human cells but become elevated in tumor cells (in which isocitrate dehydrogenases are mutated), the authors focus on the impact of loss or mutation of the S. cerevisiae ortholog IDP2, as well as loss of DLD2, a D-2 HG dehydrogenase. Using the CRASH assay to detect changes in gene expression at a reporter locus, the authors focus on inhibition of histone demethylation without the confounding influence of DNA methylation/demethylation. The authors conclude that the demethylases Rph1 and Gis1 are inhibited by D2-HG, leading to increased H3K36 methylation and gene repression at this locus.*

*While these results are of interest, they do not appear to unveil new biological insights regarding histone demethylation and/or silencing to merit inclusion in eLife. Unfortunately the authors do not revisit the IDP2 mutation in human tissue culture and thus, in its current form, this work does not significantly advance our understanding of histone demethylation, D-2 HG regulation in yeast, gene silencing or how this mechanism functions to inhibit histone demethylation in the context of human disease. While the work itself is interesting, these studies do not establish a direct link between inhibition of demethylation and repression of gene silencing, leaving open the possibility of an indirect mechanism. The authors may want to consider addressing additional key questions such as when in the yeast life cycle or under what stress conditions might up-regulation of D-2 HG result in a meaningful biological outcome with respect to histone methylation or gene silencing.*

We admire the reviewer’s expansive view of the relevance of our work and his/her desire to see it replicated in human tissue culture cells. Indeed, we are anxious to see the work replicated in bona fide tumor cells and made multiple inquiries including through Genentech and UCLA to see whether we could wrap such investigations into this study. Without going through the convoluted narrative of these investigations, in the end we resolved to play to our strength, confident that the robust cancer research community will quickly follow up.

To emphasize the importance of this work on its own, without mammalian data, we have rigorously tested and demonstrated one leading hypothesis for the mechanism by which the archetypal oncometabolite D2-HG exerts its carcinogenetic effect. Accumulation of D2-HG in certain cancers leads to hypermethylation of DNA and histones and altered gene expression. In these cancers, the link between the inhibition of DNA demethylation and cancer are, while indirect, compelling: these cancers have either mutations in IDH or mutations in the TET family of demethylases, but not both. While most studies to date on D2-HG have pointed towards the model that D2-HG alters the state of chromatin through dysregulation of an epigenetic process, there are other classes of cancer in which IDH mutations are drivers, but for which TET mutations are not found, and hence the conceptual link is not closed. Our work provides a compelling way to think about those cancers.

We utilized a unique genetic tool (the CRASH assay), only available in *Saccharomyces*, to study the effect of D2-HG on heterochromatin and gene silencing. We unequivocally demonstrated a mechanism by which D2-HG influences chromatin structure and dynamics, we demonstrated it occurs in a way that results in a change in gene expression through interference of an epigenetic mechanism, and we demonstrate that is absolutely independent of any influence on DNA methylation of demethylation. This is the first such mechanistic demonstration of the consequence of oncometabolite inhibition of histone demethylases with compelling causality. Moreover, we have demonstrated new aspects of D2-HG metabolism; the function of Dld2 and Dld3 are differentially regulated based on the mode of carbon metabolism and energy production in the cell (oxidative phosphorylation versus fermentation). We also demonstrate a novel and unanticipated connection between methylation of H3K36 and silencing.

The reviewer touches on important questions that remain regarding the cellular conditions under which D2-HG is produced. Those questions are of great significance and are the subject of extensive on-going experiments in the lab, including but not limited to, identifying other targets of D2-HG affecting cellular function. We hope to be lucky enough to engage this reviewer on our next submissions.

*In addition, there are several major and minor concerns to address:*

*1) One significant problem is that the D-2 HG levels do not increase for the IDP2-R132H mutant when compared to wild type using LC-MS in Figure 3, contrasting with the CRASH results in Figure 1. The authors attempt to address this apparent discrepancy by transforming with a plasmid with native promoter-driven IDP2 (Figure 3), but they do not then revisit the CRASH assay with plasmid-based IDP2 overexpression.*

In Figure 3 we demonstrated a single copy of the *IDP2-R132H* mutation did not result in increased D2-HG levels measured by mass spectrometry. Tumors with the equivalent isocitrate dehydrogenase mutation (*IDH1-R132H*) are exclusively heterozygous, retaining a wild-type copy of *IDH1*. Moreover, in tumor cells knock-down of wild-type *IDH1* reduces D2-HG produced by *IDH1-R132H*. We wondered if the yeast enzyme Idp2-R132H might also depend on the presence of wild-type Idp2 and this was specifically tested in Figure 3. We transformed *IDP2-R132H* cells with a plasmid containing a wild-type *IDP2* gene expressed from its native promoter and observed that the presence of a wild-type copy of *IDP2* in combination with the *IDP2-R132H* mutation resulted in detectable increases of D2-HG. The reviewer points out that our model would be considerably strengthened if we also observed effects on silencing that correspond to this increased D2-HG production that we measured by mass spectrometry. We agree with the assessment of the reviewer, and we have now included this analysis. The primary question at hand was whether the additional D2-HG that results from a copy of plasmid-borne wild-type *IDP2* in an *IDP2-R132H* strain also results in increased gene silencing. Indeed, we saw an increase in silencing in *IDP2-R132H* strains that contained an additional copy of the wild-type *IDP2* gene compared to *IDP2-R132H* strains containing a control plasmid. Like the production of D2-HG, this effect on silencing was not simply due to an extra copy of wild-type *IDP2* (no effect on D2-HG levels or silencing were observed in strains with two copies of wild-type *IDP2*), but rather specific to the combination of IDP2-R132H with a copy of wild-type *IDP2*. These data were added as Figure 3—figure supplement 1, and they greatly support and strengthen our conclusions that a wild-type copy of *IDP2* stimulates the production of D2-HG by *IDP2-R132H*.

*Furthermore, Figure 3 lacks error bars. Since the Materials and methods section does not describe any plasmids used in the paper, plasmid copy number is not addressed.*

Thanks for catching that. We have added error bars (Standard Error of the Mean) to Figure 3 graphs, the differences were assessed with statistical testing (detailed in the text and figure legend), and the description of the construction of the plasmids was added to the methods section. The plasmid is centromere-based and expresses *IDP2* from its native promoter. Therefore this plasmid is typically maintained as a single copy and expression of *IDP2* behaves as the native locus.

*2) The immunoblot quantifications in Figure 6 should be normalized to H3 in some manner – preferably using LI-COR to probe for H3 methyl marks and H3 on the same membrane.*

As suggested, we have requantified the immunoblots in Figure 6 by normalizing to H3 levels. A description of the quantification is included in the Figure 6 legend. While this has not changed the take-home message of the experiment, the clarity of the conclusions has clearly been enhanced.

*3) The authors state that there is no H3K79 demethylase in S. cerevisiae, which is a much stronger statement than that none has been identified.*

The reviewer is absolutely correct. We have changed the wording to ‘no known demethylase’ to reflect this.

*4) The 2007 Genetics paper by Tompa & Madhani should be referenced.*

This was an excellent suggestion by the reviewer. The referenced work demonstrates the role of H3K36 methylation in preventing ectopic binding of the Sir proteins to heterochromatin- and telomere- adjacent regions. Importantly, this opens the possibility that increased global H3K36 methylation due to D2-HG accumulation could, in principle, prevent binding of Sir proteins to non-heterochromatic regions, making more Sir protein available to bind at heterochromatic loci. We have added this reference and incorporated the above points into a discussion of its important implications for our study. This was an important addition that deepens the discussion of our results in this manuscript.

*5) All instances of the term "Li-Core" should reflect the true name "LI-COR."*

Thank you for catching that. We have changed all instances to reflect the correct name.